# Revealing the status of *Orbicella*: Main reef-builder of Morrocoy National Park and Cuare Wildlife Refuge, Venezuela, Southern Caribbean

**Anaurora Yranzo-Duque**[1,2,3]*, **Ana Teresa Herrera-Reveles**[3,4☯], **Estrella Villamizar**[1,3☯], **Francoise Cabada-Blanco**[2,5☯], **Jeannette Pérez-Benítez**[1,3‡], **Hazael Boadas**[3‡], **José G. Rodríguez Quintal**[3,6‡], **Carlos Pereira**[7‡], **Samuel Narciso**[3,8‡], **Freddy A. Bustillos**[9‡]

**1** Laboratorio de Ecología de Sistemas Acuáticos, Línea de Investigación Ecosistemas Marino-Costeros, Centro de Ecología y Evolución, Instituto de Zoología y Ecología Tropical, Universidad Central de Venezuela, Caracas, Venezuela, **2** EDGE of Existence Programme, Conservation and Policy, Zoological Society of London, London, United Kingdom, **3** Fundación Científica Los Roques, Caracas, Venezuela, **4** Laboratorio de Ecología Humana, Centro de Ecología Aplicada, Instituto de Zoología y Ecología Tropical. Universidad Central de Venezuela, Caracas, Venezuela, **5** Institute of Marine Sciences, School of the Environment and Life Sciences University of Portsmouth, United Kingdom, **6** Laboratorio de Biología Marino Costera, Departamento de Biología, Universidad de Carabobo, Valencia, Venezuela, **7** Laboratorio de Taxonomía y Ecología de Algas y Macrofitas Marinas, Instituto de Biología Experimental, Universidad Central de Venezuela, Caracas, Venezuela, **8** Fundación para la Defensa de la Naturaleza, estado Falcón, Venezuela, **9** Gerencia de Ordenación Pesquera, Instituto Socialista para la Pesca y Acuicultura, Caracas, Venezuela

☯ These authors contributed equally to this work.
‡ These authors also contributed equally to this work.
* cursosayranzo@gmail.com

## Abstract

Reef-building corals are the main basis of coral reef ecosystems, and the *Orbicella* genus is currently the most important in the Caribbean region. Although *Orbicella* species have been extensively studied, gaps in some southern Caribbean areas still exists on their status, which is crucial for management and conservation plans. In this study, we aimed to describe the population status of *Orbicella faveolata* and *Orbicella annularis* in two coastal Marine Protected Areas (MPA) of Venezuela: Morrocoy National Park and Cuare Wildlife Refuge. Between 2018-2020, 16 reefs in five sectors were surveyed using the Atlantic and Gulf Rapid Reef Assessment Protocol. Among the two *Orbicella* species, *O. faveolata* was dominant in both MPA´s with higher densities and live cover, varying at both reefs and sectors. The central sector of Morrocoy and reefs from Cuare are the most relevant for *Orbicella* populations, due to the higher live cover and abundance of reproductive colonies. Diseases were the primary threat recorded for both species. Adequate MPA management is essential for the conservation of the Morrocoy-Cuare coral system, including the reduction of local anthropogenic stress sources, such as oil spills, uncontrolled tourism and sewage discharges.

## Introduction

The relevance of coral reefs globally is extensively documented [1,2] as is that of the coral species responsible for building most of the structural reef framework [3,4]. As such, reef-building

---

**Data availability statement:** All relevant data are within the article and its supporting Information files.

**Funding:** This work was supported by the EDGE of Existence Program of the Zoological Society of London (https://www.edgeofexistence.org/blog/meet-new-orbicella-corals-fellow-ana-yranzo/) and the Clear Reef Social Fund (https://www.clear-reef.com/social_fund-objective_of_the_fund.php), both awarded to Anaurora Yranzo Duque. The EDGE Fellowship program provided training related to the project from which this manuscript was derived. The funders had no role in study design, data collection and analysis, decision to publish, or preparation of the manuscript.

**Competing interests:** The authors have declared that no competing interests exist.

corals are key for the ecosystem function of coral reefs and therefore, for maintaining the many benefits they provide to human populations [5]. After the decline of *Acropora* species in the early 1980s due to white band disease episodes, along with other stressors [6], *Orbicella* was considered the most important genus of shallow reef building corals in the Caribbean region [7,8]. *Orbicella* species crucially contribute to reef carbonate production rate and the structural complexity provided by their colonies, enhances habitat heterogeneity [9,10]. They are distributed throughout over 30 countries in the Caribbean, including southern Florida, Bahamas, Bermuda and Gulf of Mexico. Nevertheless, their populations, especially *Orbicella annularis* and *Orbicella faveolata*, have experienced a drastic decline caused by bleaching events, disease episodes, and other negative impacts caused by anthropic activities [11,12] evidenced by a reduction in live coral cover and colony abundance [5,13]. Currently, *O. annularis* and *O. faveolata* are classified as Endangered according to the last International Union for Conservation of Nature (IUCN) Red List of Threatened species assessment [14]. They are also included in CITES (Appendix II), the SPAW Protocol (Annex III), and in 2014, the National Marine Fisheries Services from the United States (NMFS) included them under the Endangered Species Act (ESA) highlighting the need for conservation action across the species' range.

Implementing local conservation measures for endangered species requires a good understanding of the specie's status locally. In Venezuela, a southern Caribbean country, information on *Orbicella* species is limited to some areas. The most recent publication dates back to 2015, which is part of the Red Book of Venezuelan Wildlife, a National level Red List of threatened species assessment, where *O. annularis* was classified as Vulnerable and *O. faveolata* as Least Concern [15]. Here we describe the population status of *O. faveolata* and *O. annularis* during 2018-2020, in the coastal Marine Protected Areas (MPA) Morrocoy National Park (MNP) and Cuare Wildlife Refuge, the only true fully carbonate coastal reefs in the country, which are considered the most developed of the continental area of Venezuela [16]. Extensive mangrove forests, seagrass meadows, and reef patches harbor high biodiversity in both protected areas [17,18]. Surrounding communities depend directly on the MPAs ecosystems, as their main economic activities are tourism and artisanal fishing. However, restricted activities are established in the Zoning Management Plans, which includes a higher access restriction in Cuare than in Morrocoy (only some tourism activities allowed), as the Wildlife Refuge have a stricter legal basis than National Parks in the country.

In 1996, a massive die-off resulted in a 90% mortality of the benthic fauna in MNP, including a drastic drop in live coral cover and diversity [19]. The die-off is believed to have been caused by a climatic and oceanic anomaly that caused a sudden oxygen depletion or a chemical pollution [20]. Additionally, chronic anthropogenic pressures have affected MNP for decades [21], including unregulated tourism, high sedimentation, sewage discharges, and pollution from nearby oil refinery and petrochemical facilities [22,23]. Unlike Morrocoy, coral reefs in Cuare were unaffected by the massive die-off, possibly due local circulation patterns [24].

This study is the first to focus on the ecology of *Orbicella* in this area of the southern Caribbean region. We specifically assessed spatial differences in abundance and other species attributes at reef scales as well as across sectors to better inform area-based conservation measures.

## Materials and methods

### Study area and data collection

Morrocoy National Park and Cuare Wildlife Refuge are coastal Marine Protected Areas in the central-west of Venezuela (Fig 1) and the last is included at the list of the Convention on Wetlands of International Importance, known as the Ramsar Convention (RAMSAR site).

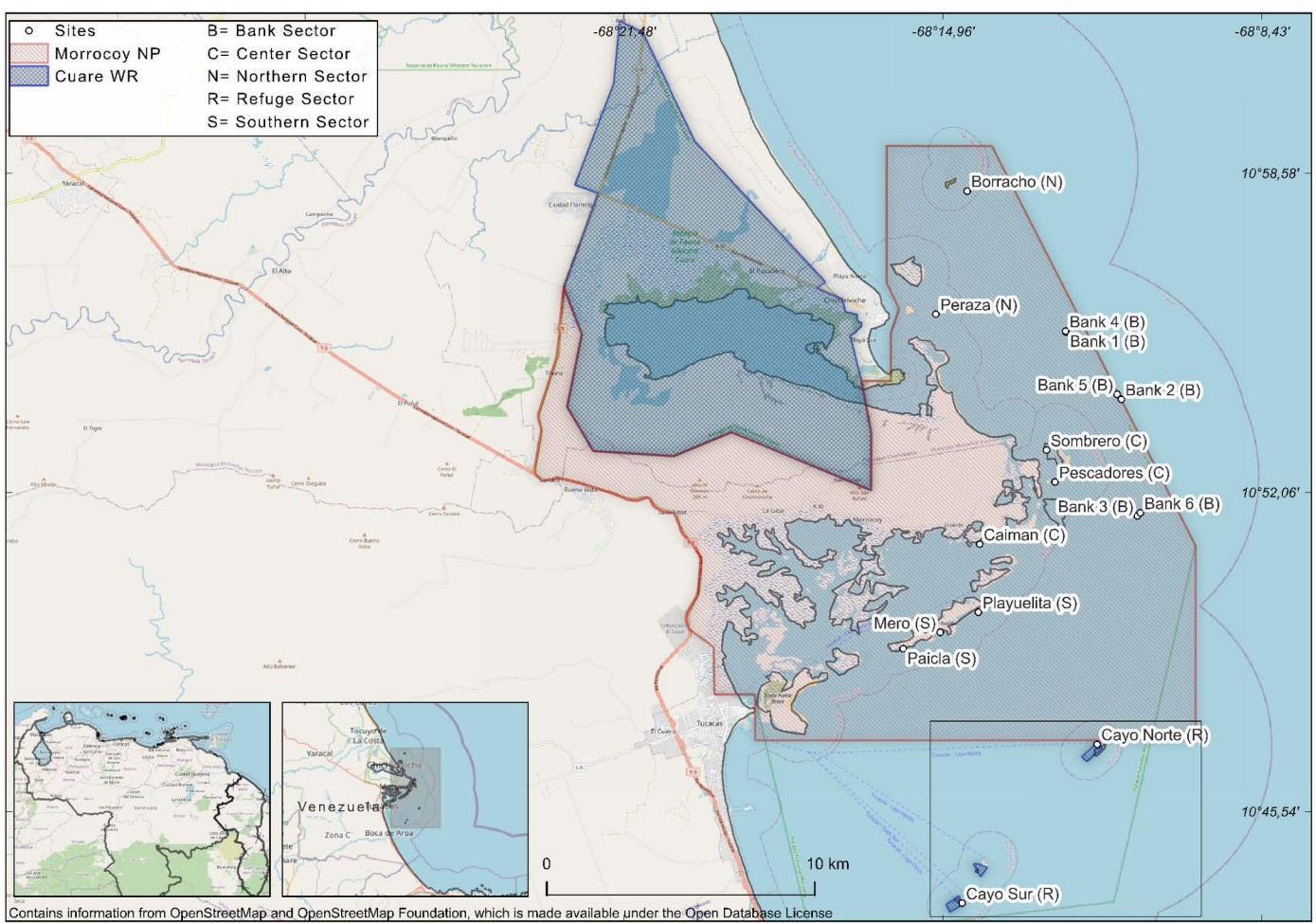

**Fig 1. Study sites in Morrocoy National Park and Cuare Wildlife Refuge, Falcón State, Venezuela, located in the Southern Caribbean.** The assessed reefs (n = 16) were classified into five sectors, indicated by the letters in parentheses next to each reef name. In the southern area of the Cuare Wildlife Refuge, only the coral cays within its boundaries are included, which are highlighted within a square frame. All sites were fringing reefs, except for the banks, which were areas dominated by branched octocorals and some sponges, along with isolated corals. These banks formed mosaics with sandy bottoms, patches of coral rubble, and rocky slabs. The base map was created using OpenStreetMap (https://www.openstreetmap.org). Transects/sector: Banks (n = 18), Center (n = 36), Northern (n = 25), Refuge (n = 24); Southern (n = 37). Transect/survey: July 2018 (n = 38), November 2018 (n = 36), July 2019 (n = 36), January 2020 (n = 30).

The status of *Orbicella* was assessed considering the following variables: density, live coral cover, size structure, and mortality in 16 reefs across five sectors (Fig 1) which were selected based on a previous distribution assessment and considered to encompass the best spatial representation of the area. A colony was defined as one entity of coral skeleton with living tissue even with separate patches of living tissue by partial mortality, but morphologically still one entity [25].

In Morrocoy National Park we assessed eight fringing reefs: two at the northern, three at the center and three at the southern of the National Park. Additionally, six banks reefs were included (Fig 1). In Cuare Wildlife Refuge, two of the three fringing reefs of this site were included.

We use a modified version of the AGRRA Protocol 5.5 (Atlantic and Gulf Rapid Reef Assessment) [26] from which a general benthos characterization was obtained. We did a

continue record of substratum cover under the transect line, instead of recording at 10 cm intervals points under the line, as the current protocol. On each reef, 10 m long x 1 m width belt transects were generally placed between 8 and 11 m depth [see 26 for further details]. Four surveys were done: July and November 2018, July 2019, and late January 2020, totaling 141 transects (see details in Fig 1). The study was approved under permit 35114 of the Ministry of Popular Power for Ecosocialism.

### Data analysis

**Live coral cover, mortality and diseases.** The data was analyzed using Primer 6 V 6.1.16 & PERMANOVA + V1.0.6. Permutational analysis of variance (PERMANOVA) was used to analyzed i) live cover of both *Orbicella* species and ii) tissue mortality categories (recent, old, and total). The PERMANOVA mixed-effect linear model was done using two fixed factors (sampling period and sector) and a random factor (reef) nested within sector based on Euclidian distance.

A contingency analysis (Pearson's chi-square) was used to assess differences in *Orbicella* diseases between sectors and survey periods. Diseases were classified according to literature gross lesion descriptions [27,28]. Disease prevalence was estimated as the number of diseased colonies/total number of colonies x 100.

Colonies were classified as alive (≤80% of partial mortality), almost dead (between >80-<100% of partial mortality), and dead (100% mortality-standing dead), following the criteria of [12].

**Colonies size structure.** Only colonies with a diameter of > 4 cm were considered, as juveniles were ≤ 4 cm in size [29]. Size frequency distribution was built for both species. For *O. annularis*, the categories of [5] were used: size class I = ≤ 50 cm²; size class II = > 50 cm²- ≤ 150 cm²; size class III = > 150 cm² - ≤ 250 cm²; size class IV = > 250 cm². For *O. faveolata* the size of the colonies was plotted considering the reproductive size reported: > 100 cm maximum diameter [30]. The reproductive size reported for *O. annularis* is > 200 cm² area according to [31]. A Permutational analysis of variance (PERMANOVA) was performed to test differences in the density of reproductive and non-reproductive colonies between sectors. In addition, an analysis of lineal regression was used to estimate the relationship between total mortality (%) and size of *O. faveolata* colonies, and an analysis of variance was used to assess the lineal regression model fits, after testing assumptions using Shapiro-Wilk normality test, Breusch-Pagan test and Durbin-Watson test.

## Results

Among the recorded coral genera, *Orbicella* had the highest relative percent cover within all sectors (Fig 2), with the exception of the banks, where *Millepora* spp. had a greater relative cover (34.81%). Live coral cover was generally dominated by *O. faveolata* (between 22.67% at the Banks and 66.63% at the Refuge), and significant spatial differences were found between sectors (F = 10.277, df = 4, p = 0.0036, CV = 35.73%) but not through time (S1 Table). The Banks and Southern sectors had the lowest cover of *O. faveolata* (t ≥ 3.00, df = 4, p < 0.05) while the highest percentage cover was recorded in the Refuge sector. There were no significant differences between the central and northern sectors (t ≤ 2.93, df = 4, p = 0.10). We found *O. annularis* in seven of the 16 reefs surveyed, including both reefs from Cuare, and in Morrocoy, two reefs of the southern, two of the center and one of the northern sectors. For this species, live cover was higher in Cuare than in the other sectors surveyed (t ≥ 3.00; df = 4, p < 0.05, average cover = 5.5%). It is important to highlight that in a shallow area of Sombrero reef (center sector) there is an extensive area of *O. annularis* colonies, unique along all the

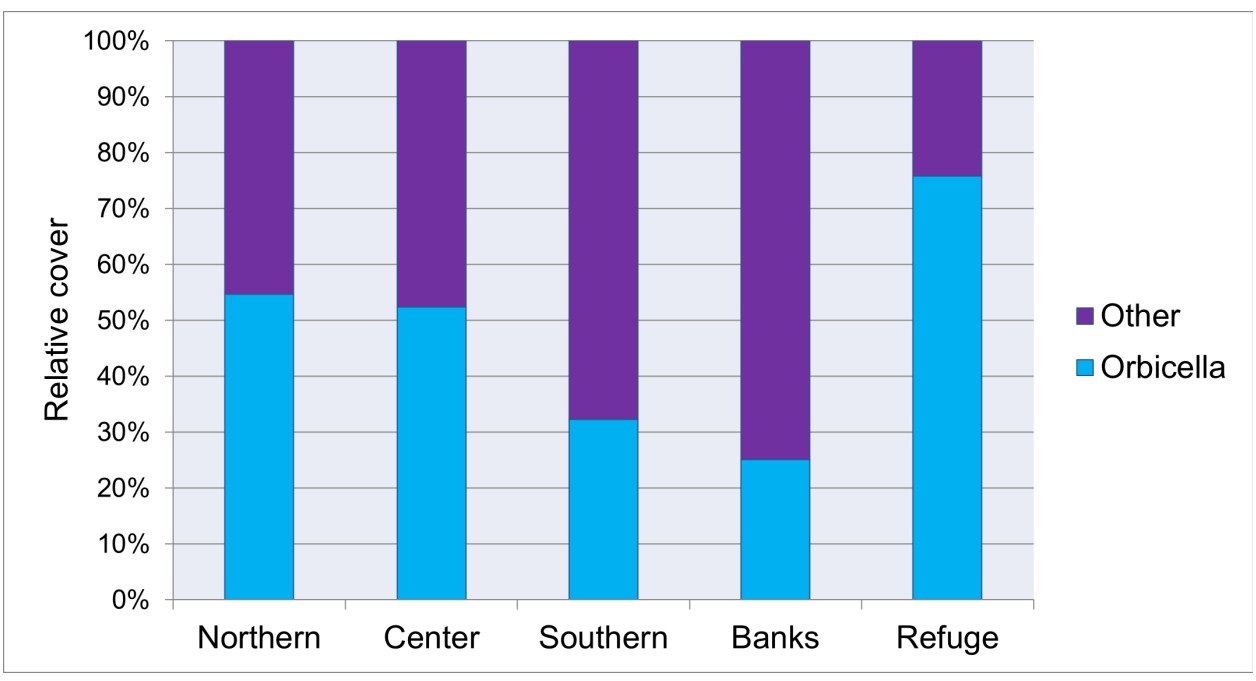

**Fig 2. Relative cover (%) of *Orbicella*: *O. annularis* and *O. faveolata*.** Study site: Morrocoy National Park (Northern, Center, Southern, Bank sectors) and Cuare Wildlife Refuge (Refuge) in 2018-2020, Venezuela, southern Caribbean.

reefs surveyed. Previous reports related to *Orbicella* live cover at the study site are summarized in S2 Table. During the study period, a total of 30 colonies for *O. annularis* and 647 for *O. faveolata* were recorded.

Concerning colony health, 85.88% of *O. faveolata* and 73.68% of *O. annularis* colonies fell within the alive category (≤ 80% of partial mortality). Average total mortality varied between 15.11% (+ 23.42) in the Southern and 35.69% (+ 22.04) in the Refuge for *O. faveolata* and 37.08% (+32.01) to 50.42% (+22.00) in the Refuge and Center for *O. annularis* (Fig 3). Tissue mortality categories showed differences between sectors (S3 Table). Old mortality stage was higher relative to recent mortality, and reefs from the Southern sector had lower percentages of recent mortality, compared to reefs in the central sector (t = 3.68, df = 4, p = 0.02) and Refuge (t = 4.64, df = 4, p = 0.01). Average values of recent mortality were between 0.45% (Banks) to 12.97% (Center) and for all sectors, recent mortality was higher in the last survey (late January 2020) than in the first one in July 2018 (t = 3.53, df = 3, p = 0.0083). For *O. annularis* there were no spatial differences in mortality categories and the average value of recent mortality for all surveys was 1.81%.

Diseases were the main detrimental factor recorded, with Yellow Band Disease being the most frequently observed in *O. faveolata* (Fig 4). The density of diseased *O. faveolata* colonies varied among the reefs ($X^2$ = 30.169, df = 4, p = 0.0015), and was lower in the Southern sector. The average prevalence of diseases varies between 2.78% in the North (Borracho reef) and 55% in the Center (Caiman reef). We recorded the highest values mostly in the Center and Refuge sectors, with an average of 25.66% (± 28.64) and 24.32% (± 22.49), respectively. In general, prevalence was highest in July with less diseased colonies found in January. However, density of the diseased colonies was not significantly different between surveys ($X^2$ = 3.425, df = 2, p = 0.1804). We found three diseases in *O. annularis*: White Plague Disease, Yellow Band disease, and Dark Spot Disease (Fig 4E-G) without differences in density of diseased colonies

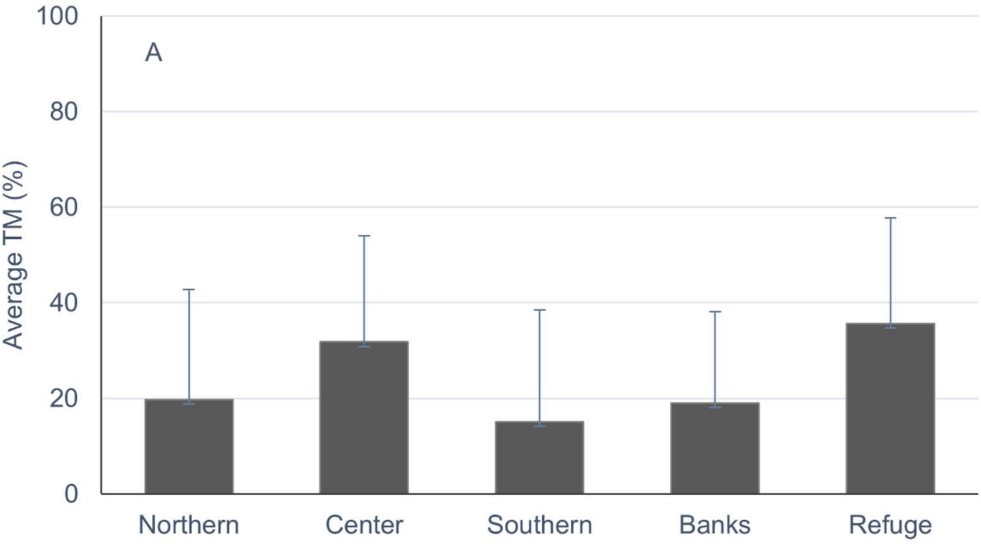

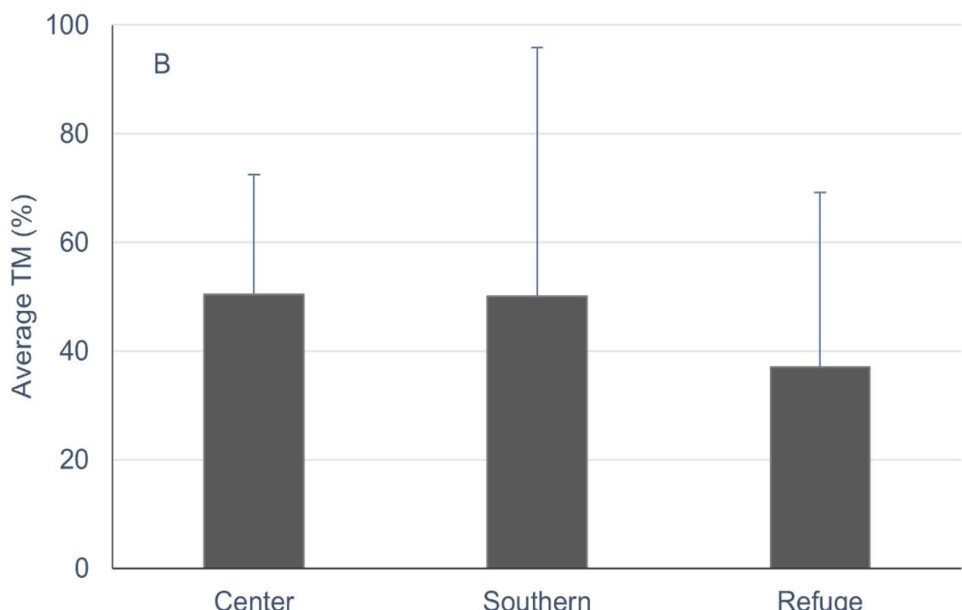

**Fig 3. Average total mortality (TM) of *O. faveolata* (A) and *O. annularis* alive colonies (B).** Study site: Morrocoy National Park (Northern, Center, Southern and Banks) and Cuare Wildlife Refuge (Venezuela). Bars are standard deviation (SD). Alive colonies = colonies with ≤80% partial mortality. Period: 2018-2020.

between reefs and sectors ($X^2$ = 4.672, df = 4, p = 0.587). We observed an additional health condition mostly in *O. faveolata* colonies, a dark line bordering the live tissue adjoining the dead area (Fig 4H). We identified other detrimental factors in fewer colonies, including bio-erosion by *Cliona* sponges, grazing by damselfish (Pomacentridae), bleaching and mechanical damage (2.44% to 8.76% of *O. faveolata* colonies and 12.5% to 25% of *O. annularis* colonies).

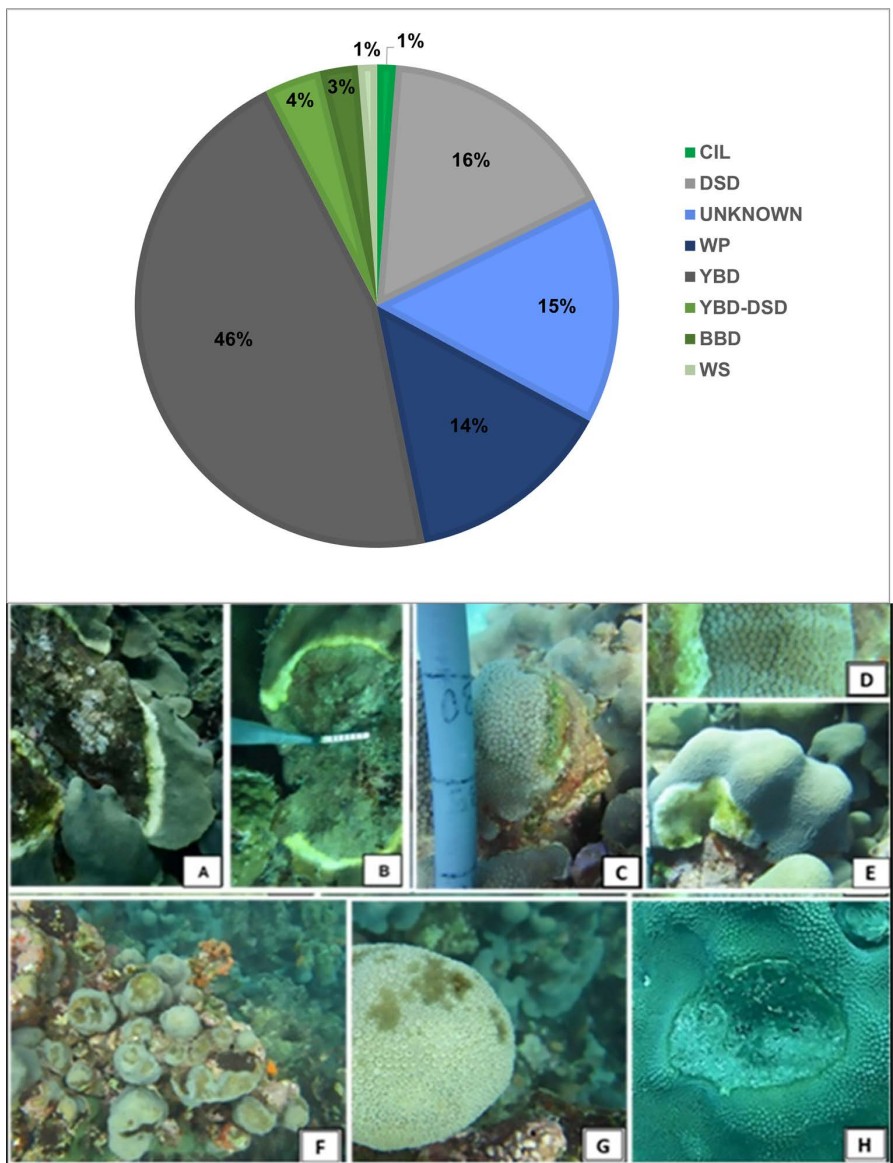

**Fig 4. *Orbicella* diseases and syndromes in Morrocoy National Park and Cuare Wildlife Refuge, in 2018-2020, Venezuela.** Graph showing diseases and syndromes in O. *faveolata*: CIL: Ciliate, DSD: Dark Spot Disease; WPD: White Plague Disease; YBD: Yellow Band Disease; BBD: Black Band Disease; WS: White Syndrome. Photos above: *Orbicella faveolata* with Yellow Band Disease (A-B); ciliate infection (C) and White Plague Disease (D); *Orbicella annularis* with White Plague Disease (E), Photos below: *O. annularis* with Dark Spot Disease (F-G) and Thin Dark Line in *O. faveolata* (H).

Of the total alive *O. faveolata* colonies measured, 42.12% were of adult reproductive size (more than 100 cm maximum diameter, Fig 5 above). Density of *O. faveolata* reproductive colonies varied among sectors (F = 7.33, p = 0.0209, %CV = 46,37, S4 Table) with higher densities in the Refuge (t ≥ 4.23, df = 4, p ≤ 0.022), followed by the Center sector (t = 1.63, df = 4, p = 0.20), where a high proportion of adult colonies was also found. Meanwhile, the Banks had densities significantly lower than those in the Northern, Central, and Refuge sectors (t ≥ 2.95, df = 4, p ≤ 0.05) and a density similar to that of the Southern sector (t = 1.44, df = 4, p = 0.24). Most of *O. annularis* colonies, were in size class IV (max. diameter = >250 cm², Fig 5 below).

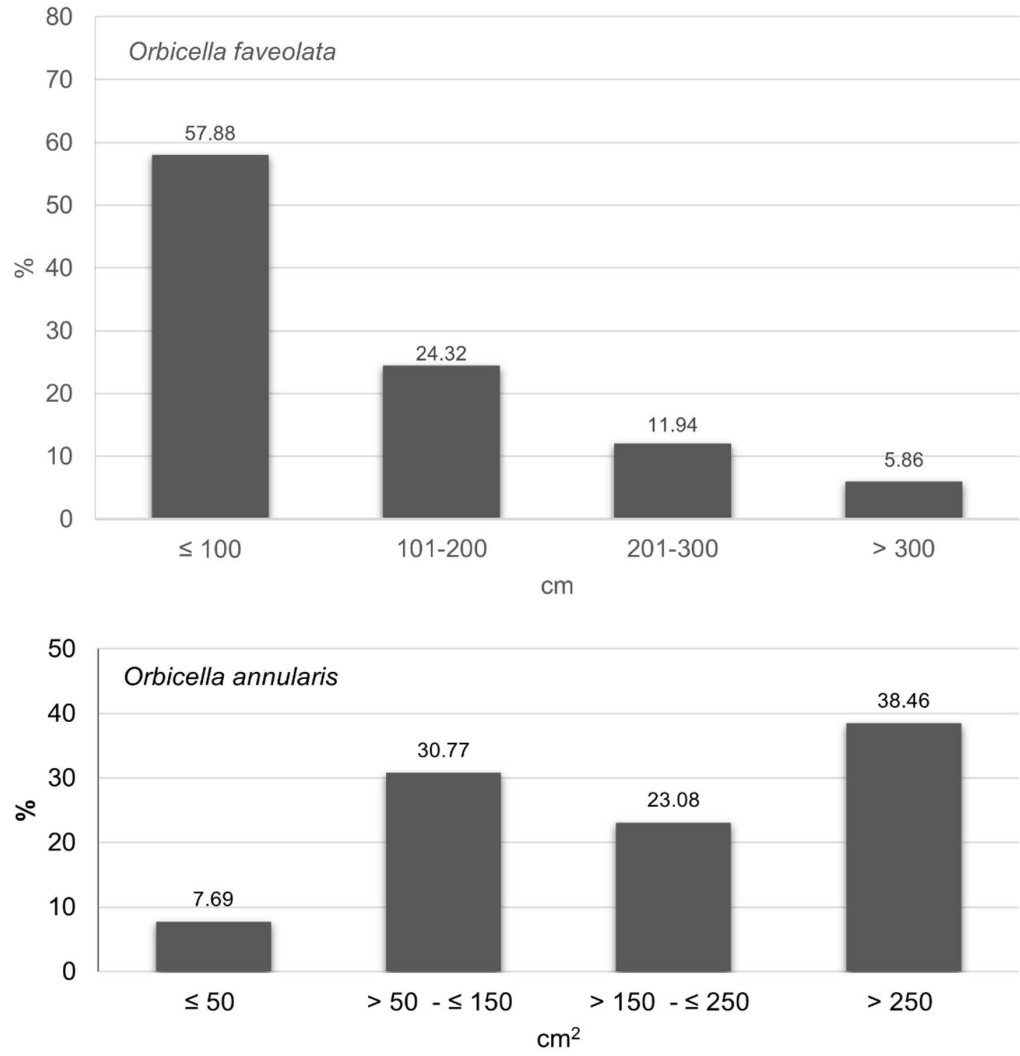

**Fig 5. Size classes in *Orbicella faveolata* and *O. annularis*.** Above, N = 444 colonies of *O. faveolata* with measures; below, N = 13 colonies of *O. annularis* with measures. from Morrocoy National Park and Cuare Wildlife Refuge (Venezuela). Period: 2018-2020. *O. faveolata* size classes according to adult reproductive sizes.

For this species, most colonies (53.84%) had a reproductive size ($> 200\,\text{cm}^2$ area) and were mainly found in Cuare (85.71%). No *Orbicella* recruits were observed during the surveys.

Finally, the simple linear regression showed a positive relationship between colony sizes and mortality for *O. faveolata* at the four periods evaluated ($R^2$ = 0.418 – 0.596, F > 0.61, p < 0.00002) (Fig 6, S5 Table). In contrast, *O. annularis* colony size and total mortality did not show a relationship ($R^2$ = 0.006, F = 0.1563, p = 0.698).

## Discussion

We assessed the status of the main coral reef builders, *O. annularis* and *O. faveolata*, in Morrocoy National Park and Cuare Wildlife Refuge in Venezuela (southern Caribbean).

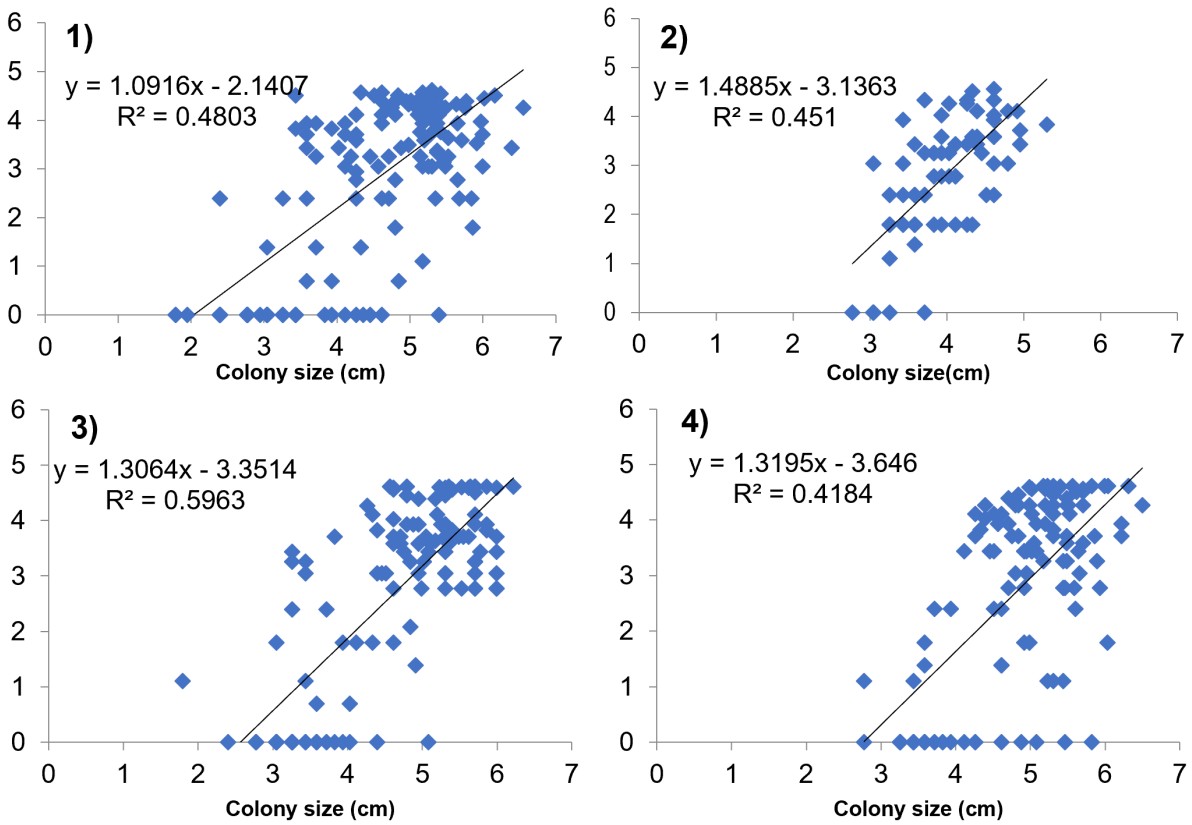

**Fig 6. Relationships between total mortality (%) and colony size (cm) of *O. faveolata* for the periods evaluated.** 1) July 2018 (n = 147), 2) November 2018 (n = 77), 3) July 2019 (n = 136) and 4) January 2020 (n = 120). Lines show the best-fit regression model and fitted functions of the form Total mortality (%) = ß0 + ß1 * colony size (cm). P values for all functions < 0.0001. R-Squared ($R^2$ = the coefficient of determination or the goodness of fit).

The results indicated that *O. faveolata* is the dominant species for coral reefs in both MPAs. Among the sectors we evaluated, the Center in Morrocoy and the Refuge in Cuare were the most relevant for *Orbicellas´*s population due to the higher live coral cover and density of reproductive colonies for both species. The reefs where we found *Orbicella* colonies are in better health, are the same reefs which were reported healthier more than 20 years ago: Sombrero at the Center sector (which had a low mortality as a consequence of the massive die-off event in 1996) and Cayo Norte in Cuare, which was unaffected by the 1996 die-off [19,24,32]. This suggests a localized effect of stressors and threats to the species within the MPA with great potential for both area-based management interventions and source-mitigation of threats to help conserve the species.

Influence of land activities modulated by local circulation has been identified as one of the main threats to Morrocoy National Park's coral reefs. The least influence of sediment discharges has been related to a better condition of reefs located at the center of Morrocoy National Park [17,24,33], unlike the northern (closest to Tocuyo River) and southern sectors (closest to Aroa river). Sombrero has the highest current speed reported for the National Park (35 cm/second) with a northwards direction [34], which avoids of the plume of sediments coming from the Tocuyo river. In case of Cuare, no direct causes have been identified, but the direction and intensity of the currents could be related [24].

Tissue mortality caused by detrimental factors among which diseases can be highlighted, is a concern for the future of *Orbicella.* Even when at the time of this study it was not observed the most recent coral disease reported: Stony Coral Tissue Loss disease (SCTLD) for which *Orbicella* species have been categorized as intermediately susceptible or highly susceptible [35], the recorded diseases are an important threat to their survival.

The dominance of *Orbicella* species before the mass mortality event in 1996 has been reported for some reefs along the Morrocoy National Park, but data was limited to live cover (see S2 Table for references) which can be a poor indicator of the abundance of a coral species, unlike colony density, especially for *Orbicella* [11]. Differences in the depth of surveys, as well as cover estimation methods (e.g., use of quadrats, chain transects) creates difficulties for obtaining a clear trend over time for *Orbicella* species. Nevertheless, taking these studies as a reference, there is a decrease in *O. annularis* live cover in most reefs, within MNP from 1999 through to 2018-2020, accounting for data in 2000, 2004 and 2005. For *O. faveolata*, there are no records of live coral cover before the massive die-off; however, considering the most recent record (years 2004-2005), the average live cover estimations for the present study are similar for some reefs. More recently in 2017–2018 [36] reported the dominance of *O. faveolata* in Caiman and Sombrero keys (center sector) as well as in the three Refuge keys (North, South and Middle).

General trends of *Orbicella* species in the Caribbean region are exhibiting a decline in live cover and abundance [8,11,37]. However, a status review [38] indicates a variable situation with mixed population trends of both reduction and stability due to the common presence of colonies in some areas (e.g., Florida, Puerto Rico, US Virgin Islands) and scarcity in others (e.g., Cuba). *Orbicella* species at the study site do not appear to have been replaced by weedy/ opportunistic corals, at least in the depth interval evaluated. Only the most deteriorated sites, e.g., the Southern sector had a higher abundance of weedy corals than *Orbicella* (e.g., Porites) which has not been the common trend within other Caribbean reefs [39,40].

The proportion of adult colonies within an area can indicate their reproductive potential of a local population. However, the effective population size may be smaller than the total standing population because of clonality and low reproductive compatibility [41]. The positive relationship between size and mortality for *O. faveolata* colonies found here, can compromise the reproductive scope of this species at the study sites, as larger colonies tend to have higher partial mortality, which is common due to more time exposed to different disturbances [42–44] and the loss of living tissue affects the ability of corals to reproduce sexually [9].Smaller colonies in the Southern sector with less mortality, are far from healthier as they are most likely a result of colony fission by partial mortality [45]. Additionally, high levels of recent mortality, like those we observed at some sites is indicative of conditions in which growth and regeneration rates are not able to keep up with mortality, and suggests that a disturbance is ongoing [46,47]. This disturbance may have increased in intensity during the last survey, when ´´red flags´´ values above 5% were recorded [48].

During the present study, no *Orbicella* recruits were observed. The only recruit recorded were of fast-growing species from the genera *Agaricia*, *Porites* and *Pseudodiploria*, which agrees with the findings of a previous study in the area [24] although other studies found recruits and juveniles of *Orbicella* [49,50]. Low recruitment of *Orbicella* genus hinders detection of recruits, making the time period of a study an important factor when assessing recruitment [5,11]. According to [29] the low recruitment of *Orbicella* might be related to high post-settlement mortality or low abundance of settlement cues from coralline algae as was found for *Acropora* [51] and not to failed reproduction or poor fertilization success [52,53]. This highlights the importance of long-term studies when assessing long-lived animals like *Orbicella.*

Mortality factors found for *Orbicella* species in the study site were in line with the literature from the Caribbean region, including Los Roques National Park in Venezuela, with Yellow Band Disease (YBD) and White Plague Disease (WPD) as the most prevalent diseases [8,54–57]. *Orbicella*'s health assessment has allowed us to obtain a broader vision of this topic as previous records of diseases from the study area were limited to Sombrero in Morrocoy and Cayo Norte in Cuare [58–61,63]. It is important to highlight that histopathology examination of diseased colonies should be done to have robust data on this topic in the study area [63,64]. With respect to the findings of the condition described as Thin Dark Line Syndrome (TDL), a temporal investigation should be performed as it could be an indicative of changes in the microbiome or in the adjacent bacterial biota of the coral (E Jordan-Dahlgren pers. comm., Nov,2022). Another hypothesis is it may be a pigmentation response to the presence of algal mats and sediments (R Rodríguez-Martínez,pers.comm., Nov,2022). Spatial differences obtained for the density of diseased colonies are probably related to density-dependent processes [65] as the reefs with fewer *Orbicella* diseased colonies had the lowest *Orbicella* abundance.

Important local stress factors not addressed like overfishing, sediment and sewage discharges, pollution including oil spills, coastal development, and associated water quality degradation are important threats in the study area, which can affect *Orbicella* species at any stage of their life cycle [66–68]. Reducing local threats is essential, as stress factors can affect coral´s immune system [69,70]. For example [62] studied the effects of hydrocarbon pollution on healthy and diseased colonies *of O. faveolata* from Sombrero reef and found reduced enzymatic activity in colonies infected with Yellow Band Disease (the most frequent disease recorded in the present study) suggesting that diseased colonies may be more vulnerable to the effects of chemical pollution. In recent years, recurrent oil spills in the area (2020,2023,2024 originating from the oil refinery located near both MPA´s [71,72] could exacerbate the detrimental effects of diseases in *Orbicellas'* within the study site. Other significant threats in the area include the invasive octocoral *Unomia stolonifera* [73] currently present in Cuare (Cayo Sur) as well as the increase in sea temperate as a result of climate change.

In conclusion, our results suggest that reefs from the Cuare Wildlife Refuge and the central sector of Morrocoy National Park should be the focus for conservation of the local *Orbicella* population to maintain and, if possible, recover the higher density of reproductive colonies and coral cover, relative to the rest of the MPA reefs. Proper management by environmental authorities, including regulating anthropic stressors, is critical for the future of these coral communities. Improving both compliance and enforcement of the current zoning with restricted access to Cuare reefs, is fundamental. Finally, responsible and sustainable tourism must be promoted and fostered in Morrocoy National Park, but especially in the central sector. Tourists and other national park users must be included in awareness raising campaigns to improve the effectiveness of the control and surveillance actions, through improved compliance.

## Supporting information

**S1 Table. *Orbicella annularis* and *Orbicella faveolata* live cover in Morrocoy National Park and Cuare Wild life Refuge, Venezuela (period 2018-2020).** PERMANOVA under a mixed-effect linear model with two fixed factors (sampling period and sector) and a random factor (reef) nested within sector based on Euclidean distance.
(DOCX)

**S2 Table. Synthesis of previous live cover reports for *Orbicella annularis* (O.ann) and *O. faveolata* (O.fav) in Morrocoy National Park (northern, center and southern sectors) and Cuare Wildlife Refuge (Refuge sector) Venezuela.**
(DOCX)

**S3 Table. Tissue mortality of *O. faveolata* colonies in Morrocoy National Park and Cuare Wildlife Refuge, Venezuela (2018-2020).** PERMANOVA under a mixed-effect linear model using two fixed factors (sampling period and sector) and a random factor (reef) nested within sector based on Euclidian distance.
(DOCX)

**S4 Table. Density of *Orbicella faveolata* colonies in reproductive and non-reproductive sizes in Morrocoy National Park and Cuare Wildlife Refuge, Venezuela (2018-2020).** PERMANOVA under a linear model with two fixed factors (sampling period and sector) and a random factor (reef) nested to the sector fixed factor.
(DOCX)

**S5 Table. Total mortality (%) and sizes of *Orbicella faveolata* colonies in Morrocoy National Park and Cuare Wildlife Refuge, Venezuela (2018-2020).** Results for the simple linear regression in each period (1-July 2018, 2-November 2018, 3-July 2019 and 4-January 2020).
(DOCX)

**S1 File. Supporting information data**
(XLSX)

## Acknowledgements

Thanks to the EDGE family for the guidance, mentoring and training during the EDGE of Existence program fellowship, especially to Olivia, Claudia, Davi, Fran, Charlie and Cassandra. Thank you to Idea Wild for the donation of essential equipment for the development of the Project and FONPESCA for the motorboat oil donation. This study would not have been possible without the teamwork of all participants. We also want to thank M. E Grillet for the valuable comments to improve the manuscript and Mirian J Davila A. for the map figure.

## Author contributions

**Conceptualization:** Anaurora Yranzo Duque, Ana Teresa Herrera-Reveles, Estrella Villamizar, Francoise Cabada-Blanco.

**Data curation:** Anaurora Yranzo Duque, Ana Teresa Herrera-Reveles, Estrella Villamizar, Francoise Cabada-Blanco.

**Formal analysis:** Ana Teresa Herrera-Reveles.

**Funding acquisition:** Anaurora Yranzo Duque.

**Investigation:** Anaurora Yranzo Duque, Ana Teresa Herrera-Reveles, Estrella Villamizar, Jeannette Pérez-Benítez, Hazael Boadas, José G Rodríguez-Quintal, Carlos Pereira, Samuel Narciso, Freddy A. Bustillos.

**Methodology:** Anaurora Yranzo Duque, Ana Teresa Herrera-Reveles, Estrella Villamizar, Francoise Cabada-Blanco.

**Project administration:** Anaurora Yranzo Duque.

**Resources:** Anaurora Yranzo Duque, Estrella Villamizar, Jeannette Pérez-Benítez, Hazael Boadas, José G Rodríguez-Quintal, Carlos Pereira, Samuel Narciso, Freddy A. Bustillos.

**Software:** Ana Teresa Herrera-Reveles.

**Supervision:** Estrella Villamizar, Francoise Cabada-Blanco.

**Validation:** Anaurora Yranzo Duque, Estrella Villamizar, Francoise Cabada-Blanco.

**Visualization:** Anaurora Yranzo Duque, Francoise Cabada-Blanco.

**Writing – original draft:** Anaurora Yranzo Duque, Ana Teresa Herrera-Reveles, Estrella Villamizar, Francoise Cabada-Blanco.

**Writing – review & editing:** Anaurora Yranzo Duque, Ana Teresa Herrera-Reveles, Estrella Villamizar, Francoise Cabada-Blanco, Jeannette Pérez-Benítez, Hazael Boadas, José G Rodríguez-Quintal, Carlos Pereira, Samuel Narciso, Freddy A. Bustillos.

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
