## [Decision Letter · Decision Letter 0]

29 Oct 2024

PONE-D-24-34496Revealing the status of Orbicella: Main reef-builder of Morrocoy National Park and Cuare Wildlife Refuge, Venezuela, Southern CaribbeanPLOS ONE

Dear Dr. Yranzo Duque,

Thank you for submitting your manuscript to PLOS ONE. After careful consideration, we feel that it has merit but does not fully meet PLOS ONE’s publication criteria as it currently stands. Therefore, we invite you to submit a revised version of the manuscript that addresses the points raised during the review process.

 **The three reviewer were generally positive in their evaluation of the manuscript. However, all of them suggested several points of attention and required the addition of more details, which the authors should carefuly consider. Moreover, a deep proofreading and polishing of the English, together with a revision of the text to increase its consistency is required.**

We look forward to receiving your revised manuscript.

Kind regards,

Erik Caroselli

Academic Editor

PLOS ONE

**Journal Requirements:**

AYD

WITHOUT NUMBER

EDGE of Existence Program of the Zoological Society of London-https://www.edgeofexistence.org/blog/meet-new-orbicella-corals-fellow-ana-yranzo/

Clear Reef Social Fund- https://www.clear-reef.com/social_fund-objective_of_the_fund.php

- The Edge Fellowship program includes training for doing the project from wich the manuscript was done

6. We note that Figure 1 in your submission contain map images which may be copyrighted. All PLOS content is published under the Creative Commons Attribution License (CC BY 4.0), which means that the manuscript, images, and Supporting Information files will be freely available online, and any third party is permitted to access, download, copy, distribute, and use these materials in any way, even commercially, with proper attribution. For these reasons, we cannot publish previously copyrighted maps or satellite images created using proprietary data, such as Google software (Google Maps, Street View, and Earth). For more information, see our copyright guidelines: http://journals.plos.org/plosone/s/licenses-and-copyright.

We require you to either present written permission from the copyright holder to publish these figures specifically under the CC BY 4.0 license, or remove the figures from your submission:

Reviewers' comments:

Reviewer's Responses to Questions

**Comments to the Author**

1. Is the manuscript technically sound, and do the data support the conclusions?

Reviewer #1: Yes

Reviewer #2: Yes

Reviewer #3: Yes

2. Has the statistical analysis been performed appropriately and rigorously? 

Reviewer #1: Yes

Reviewer #2: Yes

Reviewer #3: Yes

3. Have the authors made all data underlying the findings in their manuscript fully available?

Reviewer #1: No

Reviewer #2: Yes

Reviewer #3: No

4. Is the manuscript presented in an intelligible fashion and written in standard English?

Reviewer #1: Yes

Reviewer #2: Yes

Reviewer #3: Yes

5. Review Comments to the Author

**Reviewer #1: ** Referee to:

Revealing the Status of Orbicella: Main Reef-Builder of Morrocoy National Park and Cuare Wildlife Refuge, Venezuela, Southern Caribbean.

Manuscript Numer: PONE-D-24-34496

This work provides valuable insights into the current population status of two key species within the genus Orbicella: O. annularis and O. faveolata. These species have emerged as the dominant reef-builders in the Caribbean since the decline of Acropora in the 1990s. By focusing on two marine protected areas in Venezuela, the authors explored spatial differences in population dynamics, using a modified version of the AGRRA protocol to assess coral density, cover, mortality, and diseases between 2018 and 2020.

This research is important for broader conservation efforts as it helps expand our knowledge base about the health of these critical reef-building species. Given that previous studies on Orbicella have used varied methodologies, comparisons across studies can be challenging. However, the authors have done a commendable job of conservatively interpreting their findings in light of existing data.

One potential limitation of the study is the lack of attention to seasonal variation, which, while noted in the paper, could have been more thoroughly explored. Although there was no significant temporal variation among the sites, seasonal changes were observed and warrant further discussion. Additionally, I found the replication of transects at all sites to be somewhat unclear, which may impact the robustness of the results.

Finally, I would recommend that the authors seek assistance from a native English speaker to refine the language and improve the overall readability of the paper.

Additional Comments:

This study would benefit from a more detailed explanation of the study site’s background and setting, along with a more organized and intuitive Materials and Methods section. Clarifying the technical details would help ensure readers fully understand what is being tested. It is important that these methods can be repeated by anyone.

I was unable to locate the raw data collected, either in the manuscript or the Supplementary Information, which contradicts the Data Availability statement. The Supplementary Information only includes tables of statistical results, not the actual in situ data. This data should be made available, and the Data Availability statement should be updated accordingly

Further comments and suggestions can be found in the PDF.

**Reviewer #2:**  Please review the attached document, my observations are few, but I think they should be addressed. The acronyms MPA and MNP should be explained from the beginning. On the other hand, the conclusions paragraph must be based on the results obtained. Of course, it is important to make these observations, but I would suggest shortening this paragraph a bit.

**Reviewer #3: ** The manuscript has the potential to be published in PLOS ONE as it generally presents adequate information. However, it requires a review of the English language and should adhere to the Submission Guidelines of PLOS ONE. Below are my comments:

A review of the writing and punctuation throughout the document is necessary. The pages are not numbered, and the text is not double-spaced. Additionally, any spacing issues in the text should be corrected.

The citations and references do not follow the "Vancouver" style used by PLOS ONE. Furthermore, the figure legends are not inserted correctly after the first paragraph where each figure is cited.

The excessive use of the word “for” in the data analysis sections should be reduced. It is important to standardize terms such as “cover,” “live coral cover,” and “coral cover” throughout the manuscript.

After mentioning the full names of the species Orbicella faveolata and Orbicella annularis, the abbreviations “O. faveolata” and “O. annularis” should be used in the rest of the document.

It is crucial to clarify in the methodology which parameters defined the "status of Orbicella." For instance, in line 118, it states, “We assessed the following variables: density, cover, size structure, and mortality.” Improving the wording in this section would be beneficial.

In lines 120 and 121, it mentions, “We use a modified version of the AGRRA Protocol 5.5.” It is necessary to specify what this modification entails.

The information needs to be organized appropriately. For example, lines 128 and 129 in the data analysis section should be moved to the introduction as part of the study objectives. The initial paragraph in the data analysis section should also be restructured.

In lines 132 to 136, it is important to provide a more detailed description of the analyses, specifying what each one (ANOVA and PERMANOVA) was used for and for which species specifically. Although this information is present in the supplementary tables, it should also be included in the main text for clarity.

Line 137: Change “ej.” to English.

Line 146: Change “was” to “were.”

Lines 176 and 177: Include parentheses.

Line 180: Change “were” to “was” (check throughout the document).

Line 184: Change “varied” to “varies.”

In the results section, line 155, the percentage of “live coral cover” for O. faveolata should be mentioned. Additionally, in line 186, it is necessary to clarify what is meant by “prevalence.” In line 196, “lower frequency” needs to specify what it is referring to.

In lines 219 and 220, it states, “The reefs where Orbicella colonies are in better health conditions nowadays are the same healthier more than 20 years ago.” The ecological and conservation implications of this statement should be discussed.

I

n lines 242 and 243, translate “(Norte, Sur and Medio)” into English. In line 311, mention the year of the recurrent oil spills and add the corresponding citation.

It is important to provide sample sizes or the number of colonies analyzed for each year and in total. Furthermore, the databases used for the analyses should be attached as supplementary information or deposited in an appropriate public repository.

The selected statistical methods are generally appropriate, especially PERMANOVA for spatial and temporal comparisons. However, since ecological data often do not meet the assumptions of normality and homogeneity of variance, it is recommended to verify that the data fulfill these assumptions for ANOVA.

Including additional multivariate analyses, such as PCoA or CCA, could provide a deeper understanding of the status of Orbicella. If possible, conducting time series analyses could assess trends in coral health parameters, including seasonal effects.

In the results section (lines 155 to 157), it is important to include key values from the PERMANOVA analysis, such as the F-value, p-value, and the percentage of variation explained for each relevant factor.

In line 166, change (Fig. 3) and move it to the end of the sentence in line 169.

The results from Supplementary Table 4 support the claim that the density of reproductive colonies varies among sectors. Clarity could be improved by including in the text the proportion of explained variability (%CV), the F-value, and the p-value.

I hope my comments will contribute to improving the manuscript.

6. PLOS authors have the option to publish the peer review history of their article (what does this mean? ). If published, this will include your full peer review and any attached files.

**Do you want your identity to be public for this peer review?** For information about this choice, including consent withdrawal, please see our Privacy Policy .

Reviewer #1: **Yes: ** Chloe Lee

Reviewer #2: **Yes: ** Juan P. Carricart-Ganivet

Reviewer #3: No

---

## [Author Response · Author response to Decision Letter 1]

21 Dec 2024

I am sending all the documents and information requested

---

## [Editor Report · Decision Letter 1]

6 Jan 2025

Revealing the status of Orbicella: Main reef-builder of Morrocoy National Park and Cuare Wildlife Refuge, Venezuela, Southern Caribbean

PONE-D-24-34496R1

Dear Dr. Yranzo Duque,

We’re pleased to inform you that your manuscript has been judged scientifically suitable for publication and will be formally accepted for publication once it meets all outstanding technical requirements.

Kind regards,

Erik Caroselli

Academic Editor

PLOS ONE
---

## [Editor Report · Acceptance letter]

PONE-D-24-34496R1

PLOS ONE

Dear Dr. Yranzo Duque,

I'm pleased to inform you that your manuscript has been deemed suitable for publication in PLOS ONE. Congratulations! Your manuscript is now being handed over to our production team.

Kind regards,

on behalf of

Dr. Erik Caroselli

Academic Editor

PLOS ONE